# Fast Removal of Propranolol from Water by Attapulgite/Graphene Oxide Magnetic Ternary Composites

**DOI:** 10.3390/ma12060924

**Published:** 2019-03-20

**Authors:** Yuehua Deng, Yani Li, Wenjie Nie, Xiang Gao, Lei Zhang, Pengli Yang, Xiaochun Tan

**Affiliations:** 1College of Geology and Environment, Xi’an University of Science and Technology, Xi’an 710054, China; ynli2019@163.com (Y.L.); nwj@xust.edu.cn (W.N.); gaoxiang@iccas.ac.cn (X.G.); leizh1981@xust.edu.cn (L.Z.); pl18209269719@126.com (P.Y.); txc9150@163.com (X.T.); 2Shaanxi Provincial Key Laboratory of Geological Support for Coal Green Exploitation, Xi’an 710054, China

**Keywords:** attapulgite, propranolol, graphene oxide, adsorption, magnetic materials

## Abstract

In this work, a novel adsorbent attapulgite/graphene oxide magnetic composite (ATP/Fe_3_O_4_/GO) was synthesized for removing propranolol (PRO) from aqueous water. The factors affecting the PRO adsorption process onto ATP/Fe_3_O_4_/GO including pH, ionic strength, sorbent dosage, and humic acid were systematically investigated by batch experiments. Meanwhile, magnetic attapulgite (ATP/Fe_3_O_4_) and magnetic graphene oxide (GO/Fe_3_O_4_) were prepared for the comparison of the adsorption performance for PRO. The structural and surface characteristics of the resulting materials were characterized by X-ray diffraction, Fourier transform infrared spectroscopy, zeta potential measurements, and scanning electron microscope. The results showed that the adsorption rate of PRO onto ATP/Fe_3_O_4_/GO was up to 99%, faster and higher than that of other adsorbents involved at neutral pH. Moreover, the adsorption kinetics were better fitted with pseudo-first-order kinetic model than the second-order kinetic model. The adsorption data were fitted well with the Freundlich isotherm equations, implying that the adsorption process was heterogeneous. The adsorption reaction was endothermic and spontaneous according to the thermodynamic parameters. All results indicated that ATP/Fe_3_O_4_/GO was a promising adsorbent for removing PRO from water.

## 1. Introduction

Pharmaceuticals and personal care products (PPCPs) are highly consumed for people’s life quality, and there has been increasing awareness of the widespread presence of PPCPs in the environment at concentrations presenting as toxic for aquatic and soil ecosystems [1,2,3]. These drugs have low retention in the human body with little or no changes to the chemical structure. After ingestion, they are excreted to the sewer system in the form of the non-metabolized parent compound or as metabolites and end up in sewage treatment plants [4,5]. However, they cannot be removed completely by traditional sewage treatment process [6,7,8] and eventually enter the natural water body and soil [9,10]. Propranolol (PRO), as a nonselective β-blocker extensively used in medicine, is mainly used to treat cardiovascular diseases including hypertension and cardiac arrhythmias [11]. The published data showed that they had been detected worldwide, ranging from ng/L to μg/L [12,13]. Moreover, some studies suggested that PRO has caused acute and chronic harm to organisms, which has been considered as the most toxic harm [14,15]. It can even transmit through the biological chain and accumulate in organisms [16]. Hence, it is very meaningful to effectively remove PRO from the aqueous environment.

Many methods have been investigated for the removal of PRO, such as degradation [17], oxidation [18], and adsorption [19], etc. Among them, adsorption is regarded as being the most efficient, economical, and easily applicable approach. Nevertheless, traditional adsorbents have some disadvantages, such as a high cost of manufacturing, difficulty regenerating, and a lack of ease in separating from water, which limit their large-scale application to some extent. Consequently, the selection of adsorbents should be paid more attention.

Attapulgite (ATP) is one kind of clay mineral with favorable physical and chemical properties such as bedded structure, fibrous morphology, a large specific surface area, and proper cation exchange capacity, which is widely used to remove pollutants from water [20,21]. Also, ATP has a high affinity to positively charged contaminants because of the permanent negative charges on its surface [22]. After adsorbing pollutants, however, ATP cannot easily be separated from water, which may cause secondary migration of the adsorbed substances or secondary pollution. In order to solve this problem, Fe_3_O_4_ particles have been loaded onto the surface of ATP, which can be separated from the medium by a simple magnetic process [23,24,25]. However, Fe_3_O_4_ particles are difficult to be well-distributed on the surface of ATP due to the fact that ATP with a large proportion is hard to suspend uniformly in solvents [26]. Furthermore, the adsorption capacity of the magnetic ATP becomes lower than that of ATP. Therefore, a suitable dispersant with a high adsorption ability for PRO is needed to improve the overall adsorption properties of the as-made material mentioned above.

In recent years, graphene oxide (GO) has become a hot spot in the field of the environmental functional materials. Due to the large surface area and special lamellar structure of GO, it can be applied in base supporting and provide a larger number of active adsorption sites for the pollutants [27]. It was reported that PRO can be efficiently removed by GO with a number of oxygen-containing functional groups on the surface of GO, which can facilitate the interaction with the target by hydrogen bonding and by π–π and electrostatic interactions [15,28,29,30,31]. Taking above results into consideration, GO may be used not only as a supporter for dispersing ATP but also as a coadsorbent to remove PRO. Up to date, there has been no reports in the literature of the adsorptive removal of PRO from an aqueous solution by an ATP/Fe_3_O_4_/GO ternary magnetic composite material, which can be easily separated with an external magnetic field and has a much higher adsorption capacity.

In this work, the ATP/Fe_3_O_4_/GO ternary composite was synthesized and applied as an adsorbent to remove PRO from aqueous solutions. The factors affecting PRO adsorption were systemically investigated, and the adsorption kinetics and adsorption isotherms for PRO were studied in detail. Several proposed mechanisms for adsorption were also discussed.

## 2. Materials and Methods

### 2.1. Materials

The ATP used in this work was obtained from Xuyi, Jiangsu, China. PRO (Energy Chemical, Shanghai, China) was used as received, and humic acid (HA) (Aladdin, Shanghai, China) was used without any further purification. Graphite powder was purchased from Guangdong Xilong Chem. Co. Ltd., Shantou, China. The methanol and acetonitrile (Tedia, Fairfield, OH, USA) were of HPLC grade. The other chemicals used were of analytical grade. All solutions were prepared using twice-distilled water. The formic acid buffer (pH 2.8) was freshly prepared by dissolving 1 mL of formic acid in 1 L of water.

### 2.2. Materials Preparation

GO was synthesized with the modified Hummers method [32]. Potassium persulfate (1 g), phosphorus pentoxide (1 g), and graphite power (1 g) were stirred to 10 mL of concentrated sulfuric acid and reacted at 80 °C for 5 h. It was then filtered, washed with distilled water until neutral pH, and dried at 60 °C in a vacuum drying oven. Concentrated sulfuric acid (40 mL) was added to the above product at a low temperature. Then potassium permanganate (4 g) was added to the solution. After the removal of the ice-bath, the reaction was carried out at 35 °C for 2 h. Distilled water (100 mL) was slowly added into the mixture and kept the temperature at 98 °C for 15 min. Then, hydrogen peroxide was added until the solution was golden yellow, and the solution was further diluted with distilled water. After being settled, the supernatant was decanted and the precipitant was washed with hydrochloric acid and distilled water until a neutral pH. Finally, the mixture dried at 60 °C and passed through a 200-mesh sieve.

Then, 100 mL of a water solution with 0.5 g of ATP and 0.5 g of GO were sonicated for 30 min and then bubbled N_2_ into the solution for 15 min. After that, the temperature was kept at 90 °C, and ferrous sulfate heptahydrate (2 g) was added into the above solution with N_2_ bubbling. This was marked as solution A. Sodium hydroxide (1.8 g) and sodium nitrate (0.9 g) were dissolved in distilled water (40 mL), and the solution was marked as solution B. Solution B was added to solution A drop by drop under nitrogen bubbling. Then, the mixture was kept for 4 h at 90 °C. After cooling, it was washed until a neutral pH, dried at 60 °C, and passed through a 100-mesh sieve. ATP/Fe_3_O_4_ and GO/Fe_3_O_4_ were also prepared based on the abovementioned procedures without the addition of GO and ATP, respectively.

### 2.3. Batch Experiment

The experiments were conducted using 40 mL glass vials at room temperature. The pH of the solution was adjusted by adding 0.1 mol/L HCl or 0.1 mol/L NaOH solutions. For a comparison of the adsorption capacities of PRO onto all adsorbents, 0.01 g of ATP, GO, ATP/Fe_3_O_4_, GO/Fe_3_O_4_, and ATP/Fe_3_O_4_/GO were added into the PRO solution given initial concentrations, respectively. The pH value was adjusted to 3, 7, and 11. Then, they were shaken for 24 h to reach equilibrium. The effects of experimental parameters of pH (3–12), dosage (0.001–0.04g), ionic strength (0–500 mg/L), humic acid concentration (50–400 mg/L), contact time (15–960 min), initial PRO concentration (5–250 mg/L), and temperature (25 °C, 35 °C, and 45 °C) on the removal of PRO were investigated in a batch technique of operation. At the end of the adsorption, the solution was filtered through a 0.22 μm membrane filter and then analyzed by HPLC. The adsorption rate and adsorption capacity were calculated to determine the adsorption properties of the materials.

### 2.4. Characterization Techniques

Chromatographic separations were obtained using a 4.6 × 150 mm SB-C18 column (Agilent, Santa Clara, CA, USA). The analytical wavelength was set at 290 nm, and samples of 40 μL were injected into the HPLC system. The mobile phases were 0.1% formic acid-acetonitrile and acetonitrile in the ratio of 70:30 (*v*/*v*) at a flow rate of 1.0 mL/min. The quantity of adsorbed PRO was calculated from the decrease of the PRO concentration in solution.

The X-ray diffraction (XRD) patterns of the samples were recorded on an X-ray diffractometer (ARL Co., Switzerland) with Cu Ka radiation from 5° to 90° (2θ). Fourier transform infrared spectra (FTIR) were used to detect the surface functional groups by a FTIR spectrophotometer (Thermo Electron Nicolet-360, USA) using the KBr wafer technique. The zeta potentials of the materials were determined on a zeta potential analyzer (Brookhaven Instruments Corp., USA). Scanning electron microscopy (SEM) images were also taken with accelerating voltage of 15.00 kV (Japan Electronics Corporation).

## 3. Results

### 3.1. Material Characterization

#### 3.1.1. FTIR Spectra Analysis

The FTIR spectra of ATP, GO, ATP/Fe_3_O_4_, and ATP/Fe_3_O_4_/GO are shown in Figure 1. The spectrum of ATP showed peaks at 3620 cm^−1^ that represented the stretching vibrations of Al–OH, and the peaks at 3432 cm^−1^ and 1635 cm^−1^ were attributed to the bend vibration of the zeolite water. The peaks at 1031 cm^−1^ and 467 cm^−1^ were attributed to the asymmetric stretching modes of Si–O–Si, whereas the peak at 796 cm^−1^ might have corresponded to the stretching vibration of Al–O–Si [33].

Compared with ATP, the spectra of ATP/Fe_3_O_4_ showed new peaks at 566 cm^−1^ and 427 cm^−1^ that corresponded to Fe^2+^–O^2−^ and the vibrations of Fe^3+^–O^2−^ [24], suggesting the successful load of Fe_3_O_4_ onto the surface of ATP. It can be seen from the picture that the ternary complex appeared at the same characteristic peaks as ATP in the spectrum, which indicated that ATP/Fe_3_O_4_/GO was based on ATP. The peak at 431 cm^−1^ provided evidence for the presence of Fe_3_O_4_ in the composites. The new absorption peak of the ternary composite at 1566 cm^−1^ might be ascribed to the overlap of the C=C from unoxidized sp^2^ C–C bonds at 1624 cm^−1^ on the GO and the vibration of the zeolite water at 1635 cm^−1^ on the ATP. The results indicated the presence of GO in the ternary composite. The interaction among GO, ATP, and Fe_3_O_4_ was affirmed by the change of the peaks’ intensities at about 3432 cm^−1^, 1721 cm^−1^, and 1222 cm^−1^, and these peaks of ATP/Fe_3_O_4_/GO were weaker than those of GO but stronger than those of ATP/Fe_3_O_4_. The peaks at 1721 cm^−1^ and 1222 cm^−1^ of GO were assigned to the C=O stretching in carboxylic acid and the C–O stretching in epoxy, respectively. However, ATP had no obvious characteristic peaks at peak positions, leading to the weakening of the peak intensity of the ternary complex at the above peak positions. The change in the peak intensity at 3432 cm^−1^ might be ascribed to the overlap of the O–H stretching vibrations of GO and the bend vibration of the zeolite water of ATP.

#### 3.1.2. Zeta Potential Analysis

The surface charge of the adsorbents was affected by the solution pH to a large extent. The zeta potentials of ATP, GO, ATP/Fe_3_O_4_, GO/Fe_3_O_4_, and ATP/Fe_3_O_4_/GO were determined at different pH values (Figure 2). The zeta potentials of ATP and GO were charged negatively on their surface in the determining pH range, and both ATP and GO had more negative charges with pH increase. Therefore, ATP and GO were expected to have good absorption abilities to the cationic compounds. Fe_3_O_4_ and ATP/Fe_3_O_4_ showed a higher zeta potential than the three other materials, suggesting Fe_3_O_4_ particles may change the surface charge of materials. When pH < 3.5, the zeta potential of ATP/Fe_3_O_4_/GO was close to zero, which also showed Fe_3_O_4_ played an important role in the surface properties of materials. Furthermore, when pH > 4, the zeta potential of ternary composite ATP/Fe_3_O_4_/GO was negative between GO and ATP, indicating the successful combination of ATP and GO and the feasibility of PRO removing.

#### 3.1.3. XRD Analysis

The XRD patterns of ATP, GO, ATP/Fe_3_O_4_, and ATP/Fe_3_O_4_/GO are shown in Figure 3. The characteristic peak at 2θ values of 26.6° was found in the XRD pattern of ATP. This peak was also well-matched with ATP/Fe_3_O_4_ and ATP/Fe_3_O_4_/GO, indicating the composites contained ATP. Six characteristic peaks corresponding to the Fe_3_O_4_ particles (30.06°, 35.48°, 43.16°, 53.56°, 57.08°, and 62.66°) were also observed for the ATP/Fe_3_O_4_ and ATP/Fe_3_O_4_/GO, and the peak location after modification did not shifted, indicating the Fe_3_O_4_ particles were successfully loaded onto the raw materials and that the modification might be on the surface. The peak at 42.4° was the characteristic peak of GO, which also presented at the ternary composite. The GO was well-attached to the ATP/Fe_3_O_4_.

#### 3.1.4. SEM Micrographs

SEM was used to characterize the synthesized GO and ATP/Fe_3_O_4_/GO. As it can be seen in Figure 4a, GO formed the sheet-like structure and had a smooth surface without holes on its surface. Partial agglomeration was also observed. The smooth surface of GO became coarse and dense after the loading of ATP and Fe_3_O_4_ particles (Figure 4b). Many rod-shaped ATPs were crosslinked on the surface of the GO, indicating that GO acted as a dispersant. Fe_3_O_4_ particles were also observed. They were agglomerated on the surface of GO and ATP, and the results were consistent with FTIR and XRD characterization. It was speculated that a higher increase in the surface area of the ternary composites than that of GO is reasonable, and its adsorption capacity is discussed in the subsequent sections.

### 3.2. Comparison of the Adsorption Capacities of PRO onto All Adsorbent

The adsorption rate of PRO onto all adsorbents was shown in Figure 5. It can be found that the adsorption rate of ATP was relatively low at all tested pH values. The adsorption rate of ATP/Fe_3_O_4_ was the lowest among adsorbents involved, which indicated that the loading of Fe_3_O_4_ led to a decrease of the adsorption rate of PRO onto ATP under the same quality conditions. The adsorption rate of GO had reached the highest value at pH = 3 and decreased with increasing pH value, manifesting an excellent adsorption rate to PRO. Although GO had a better removal rate under acidic conditions, it limited the application of GO in natural waters. It can be seen from Figure 5, the adsorption rate of ATP/Fe_3_O_4_/GO to PRO was about 55% at pH = 3, close to 100% under neutral conditions, and slightly decreased when pH = 11. The results suggested that GO as a dispersant can improve the adsorption rate of ATP/Fe_3_O_4_ and that ATP/Fe_3_O_4_/GO has a good performance to removal PRO from water. Therefore, ATP/Fe_3_O_4_/GO was chosen for this study.

### 3.3. Effect Factors

#### 3.3.1. Effect of pH

The pH value played an important role in the adsorption behavior. The effect of pH on PRO onto ATP/Fe_3_O_4_/GO is presented in Figure 6. It was shown that the adsorption rate of PRO increased with an increasing pH when the pH was below about 5.5. At a low pH, the H^+^ in solution competed with PRO for adsorption sites because of a strong concentration of H^+^, resulting in a lower adsorption rate of PRO. Although GO showed a good performance for PRO removal at pH = 3, the competition adsorption played a dominant role in the adsorption process. With increasing pH, the effect of H^+^ became weakened and the adsorption rate of PRO increased. In addition, PRO had secondary amines with pKa larger than 9.0 and, thus, presented primarily in a positively charged format in solution, while ATP/Fe_3_O_4_/GO exhibited a negative charge at all tested pH (Figure 2). It was also one of the likely reasons for the increase of adsorption rate that the negatively charged –COO^−^ on the surface of GO attracted the protonated –NH^3+^ of PRO. The electrostatic attraction also occurred between ATP and PRO.

When 5.5 < pH < 10, a platform had the highest adsorption rate of PRO. Combined with Figure 5, the removal rate of PRO onto ATP and GO had significantly increased and slightly decreased, respectively, which presented a small change of adsorption rate at 5.5 < pH < 10. The number of negative charges on the adsorptive sites of the ATP/Fe_3_O_4_/GO surface increased and that of positive charges on PRO decreased, but the adsorptive behavior was still influenced by electrostatic attraction. Electrostatic attraction still took part in the adsorption process in spite of a lower H^+^ concentration in this pH range. Furthermore, hydrogen bonds formed by the hydrogen and oxygen atoms of the hydroxyl group of ATP/Fe_3_O_4_/GO with the nitrogen and hydrogen atoms of the amino group of PRO may contribute to the drug removal. When pH > 10, PRO adsorption had a slow decrease ascribed to the weakness of electrostatic attraction. PRO existed as a neutral molecule due to the deprotonation of the amino group, limiting the adsorption of PRO onto ATP/Fe_3_O_4_/GO. Generally, cation exchange, electrostatic attraction, and hydrogen bonding played important roles in the adsorptive behavior of PRO onto ATP/Fe_3_O_4_/GO. It was worth mentioning that acid-activated ATP has been investigated in our previous study [22]. We found that acid-activated ATP had a similar adsorption rate to ATP/Fe_3_O_4_/GO which more easily separated from water. In general, ATP/Fe_3_O_4_/GO can maintain a high adsorption rate in a large pH range, indicating that the material can be applied into the removal of PRO in different pH water bodies.

#### 3.3.2. Effect of Adsorbent Dosage

Adsorption dosage was also an important factor for the removal of PRO from water. The effect of ATP/Fe_3_O_4_/GO mass on the adsorption behavior of PRO for a given initial concentration was exhibited in Figure 7. It can be seen that the adsorption rate decreased with the adsorption dose. At smaller dosages, adsorptive sites were occupied completely, presenting a higher adsorption rate. However, it was shown a lower adsorption rate at higher adsorption dosages. It resulted in an excess of adsorptive sites because the adsorption sites remained unsaturated during the adsorption process.

#### 3.3.3. Effect of Ionic Strength

The effect of Na^+^ and Ca^2+^ was studied by adding different concentration of NaCl and CaCl_2_ into the reaction system, respectively (Figure 8). The results showed that the PRO adsorption rate was decreased with the presence of Na^+^ and Ca^2+^. The two kinds of cations inhibited the PRO adsorption process, leading to the PRO molecules adsorbed being desorbed into the solution. The cations competed with PRO and occupied the adsorption sites on the surface of the adsorbent, resulting in only a small amount of PRO molecules contacting adsorption sites. Furthermore, the cations adsorbed onto the surface of the adsorbents and the PRO generated electrostatic repulsion, making it difficult for PRO molecules to access the adsorbent. Therefore, the inhibited PRO adsorption was ordered as Ca^2+^ > Na^+^. This suggested that cation exchange and electrostatic interaction may play important parts in the adsorption process of PRO onto ATP/Fe_3_O_4_/GO.

#### 3.3.4. Effect of Humic Acid

Humic acid (HA), a major component of dissolved organic carbon (DOM) in the natural environment, had a strong binding capacity for many contaminants and could increase the contaminant concentration in water. The effect of HA on the adsorption process of PRO is presented in Figure 9. It can be seen that the adsorption rate of PRO onto ATP/Fe_3_O_4_/GO decreased with an increasing HA concentration, which indicated that the adsorption of PRO was inhibited by HA. Compounds with the same adsorption mechanism competed for mutually available adsorption sites. HA was adsorbed onto the ATP/Fe_3_O_4_/GO and occupied the adsorption sites of the adsorbent surface. The PRO molecules could not contact the adsorption sites, leading to the inhibition of PRO adsorption. It was noteworthy that HA was charged negatively in the solution. When the HA concentration reached higher, there were a number of HAs in the solution except for HA adsorbed onto the adsorbent. Electrostatic attraction occurred between HA in solution and PRO that had been adsorbed on the surface of the adsorbent, increasing the concentration of PRO in the solution. Thus, the adsorption rate was gradually reduced.

### 3.4. Adsorption Kinetics

The kinetic data were fitted to a pseudo-first-order adsorption model (Equation (1)) and a pseudo-second-order adsorption model (Equation (2)). This was done in order to determine the time period where the adsorption process was completed.
(1)$$Ln\left(q_{e}-q_{t}\right)=Lnq_{e}-K_{1}t$$
(2)$$\frac{t}{q_{t}}=\frac{q}{K_{2}q_{e}^{2}}+\frac{t}{q_{e}}$$
where *q_e_* and *q_t_* are the amount of PRO adsorbed onto adsorbent (mg/g) at equilibrium and at time *t* (min), respectively; *K*_1_ is the rate constant of pseudo-first-order adsorption (1/min), and *K*_2_ is the constant of pseudo-second-order rate (mg/(g·min)).

The adsorption kinetics data of PRO onto ATP/Fe_3_O_4_/GO is shown in Figure 10. The fitting parameters and theory data calculated by the nonlinear equations are listed in Table 1. It can be seen that the adsorption rate was very high during the first 1 h of the process. The removal of PRO was maintained at a constant level after 1 h of contact time. As shown in Table 1, the correlation coefficient *R*^2^ for the pseudo-second-order adsorption model was relatively low, indicating that the pseudo-second-order adsorption model was a poor fit for the experimental data. However, the correlation coefficient for the pseudo-first-order adsorption model was found to be higher, suggesting that the adsorption data were well represented by pseudo-first-order kinetics, which indicated that the adsorption process might be controlled by physical adsorption.

### 3.5. Adsorption Isotherm

The Langmuir adsorption model (Equation (3)) assumed that the adsorption occurs at homogeneous sites on the adsorbent. The Freundlich adsorption model (Equation (4)) was generally used to describe the adsorption isotherm for heterogeneous surfaces.
(3)$$\frac{1}{q_{e}}=\frac{1}{q_{m}}+\frac{1}{K_{L}q_{m}C_{e}}$$
(4)$$\log q_{e}=\frac{1}{n}logC_{e}+logK_{F}$$
where *q_e_* and *C_e_* are the amounts of PRO adsorbed (mg/g) and equilibrium concentration (mg/L), respectively; *q_m_* is the maximum PRO adsorption (mg/g); *K_L_* is the Langmuir coefficient relating to the strength of sorption; *K_F_* is the Freundlich isotherm constant; and 1/*n* (dimensionless) is the heterogeneity factor.

Equilibrium isotherms data and the fitting curves for the adsorption of PRO onto ATP/Fe_3_O_4_/GO are presented in Figure 11. The fitted parameters and the theory data obtained by nonlinear equations are summarized in Table 2. It can be found that, when increasing the initial concentration of PRO, an improvement of adsorption capacity in equilibrium was observed. The adsorption data of PRO onto ATP/Fe_3_O_4_/GO was fitted with the Freundlich isotherm better than with the Langmuir model with a higher value of R^2^, indicating that the PRO adsorption was not a monolayer adsorption on the adsorbent surface. Deng et al. reported the Langmuir isotherm equation fit the adsorption isotherm data of PRO onto acid-activated ATP well [22]. Lambropoulou et al. pointed out that the adsorption process of PRO onto GO was a monolayer adsorption [15]. These results described that the adsorption process of PRO onto GO and ATP was a monolayer sorption, while the adsorption of PRO onto ATP/Fe_3_O_4_/GO was a multilayer sorption, which suggested that there may be an interaction between ATP and GO to determine the mechanism of PRO adsorption.

### 3.6. Adsorption Thermodynamics

Temperature was an important factor affecting the physical properties of PRO in aqueous solutions. Thermodynamic parameters such as Gibbs free energy change, enthalpy change and entropy change (ΔG), enthalpy change (ΔH), and entropy change (ΔS) can be calculated by the following equations:(5)$$\mathrm{Ln}\left(\frac{q_{e}}{C_{e}}\right)=\frac{\Delta S}{R}-\frac{\Delta H}{RT}$$
(6)ΔG=ΔH−TΔS
where *q_e_* and *C_e_* are the amounts of PRO adsorbed (mg/g) and equilibrium concentration (mg/L), respectively; *T* is the absolute temperature (K); and *R* is the gas constant (8.341 J/(mol·K)).

The thermodynamic data of PRO adsorption onto ATP/Fe_3_O_4_/GO are shown in Table 3. A positive value of Δ*S* indicated that the adsorption reaction was an entropy increase process. At different temperatures, a negative Δ*G* value indicated that the adsorption process of PRO onto ATP/Fe_3_O_4_/GO was spontaneous, and the adsorption reaction was adsorption reaction.

### 3.7. Comparison of ATP/Fe_3_O_4_/GO with Other Adsorbents

In order to evaluate the application potential of the prepared adsorbents in an aqueous environment, the adsorption properties of ATP/Fe_3_O_4_/GO were compared with different adsorbents in our previous study.

As shown in Table 4, the adsorption capacity of ATP/Fe_3_O_4_/GO was higher than that of Chitosan-ATP and KH550-ATP, which showed that the as-made adsorbent had a better adsorption capacity. The adsorption capacity of ATP/Fe_3_O_4_/GO was close to that of powder-activated carbon and slightly smaller than that of acid-activated ATP. The data manifested that three materials had similar adsorption capabilities, but ATP/Fe_3_O_4_/GO showed the advantage of easy separation from water. Generally, having a low cost, being nonpolluting, and having a high adsorption capacity make ATP/Fe_3_O_4_/GO an attractive adsorbent for the removal of PRO from water or wastewater.

## 4. Conclusions

In this study, ATP/Fe_3_O_4_/GO was synthesized and applied in removing PRO from water, and the results suggested that it had the best adsorption performance among all the materials involved. PRO adsorption is strongly dependent on the solution pH, adsorbent mass, coexisting cations (Na^+^ and Ca^2+^), and humic acid. The protonated –NH^3+^ of PRO was attracted by the negatively charged –COO^−^ on the surface of ATP/Fe_3_O_4_/GO, indicating that the electrostatic interactions might play an important role in PRO adsorption process. Hydrogen bonds formed by the hydrogen and oxygen atoms of the hydroxyl group of ATP/Fe_3_O_4_/GO with the nitrogen and hydrogen atoms of the amino group of PRO might contribute to PRO removal. Specially, ATP/Fe_3_O_4_/GO could maintain a high adsorption rate in the range of pH 5–11, indicating that the material had the potential to be used to remove PRO from different pH water bodies. All results of the study demonstrated that ATP/Fe_3_O_4_/GO was a promising adsorbent for removing PRO from water.

## Figures and Tables

**Figure 1 materials-12-00924-f001:**
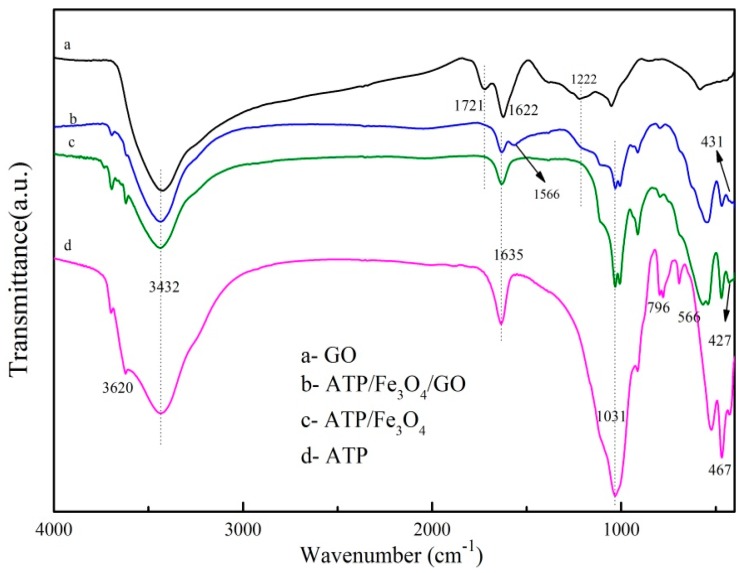
The FTIR spectra of graphene oxide (GO), Attapulgite (ATP)/Fe_3_O_4_/GO, ATP/Fe_3_O_4_, and ATP.

**Figure 2 materials-12-00924-f002:**
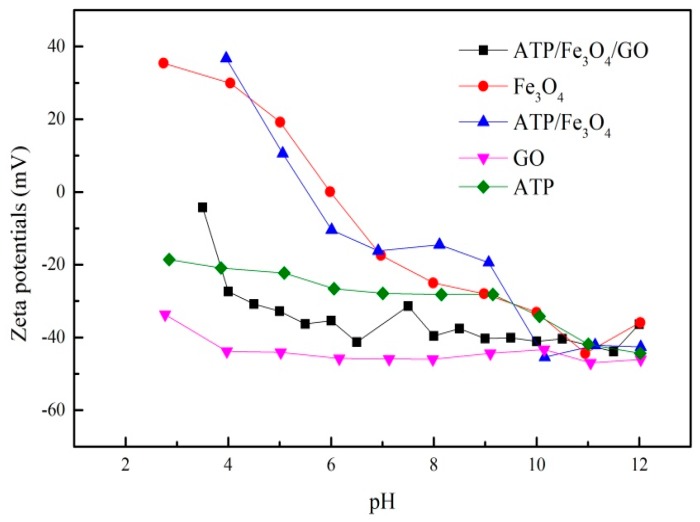
The zeta potentials of ATP, GO, Fe_3_O_4_, ATP/Fe_3_O_4_, and ATP/Fe_3_O_4_/GO.

**Figure 3 materials-12-00924-f003:**
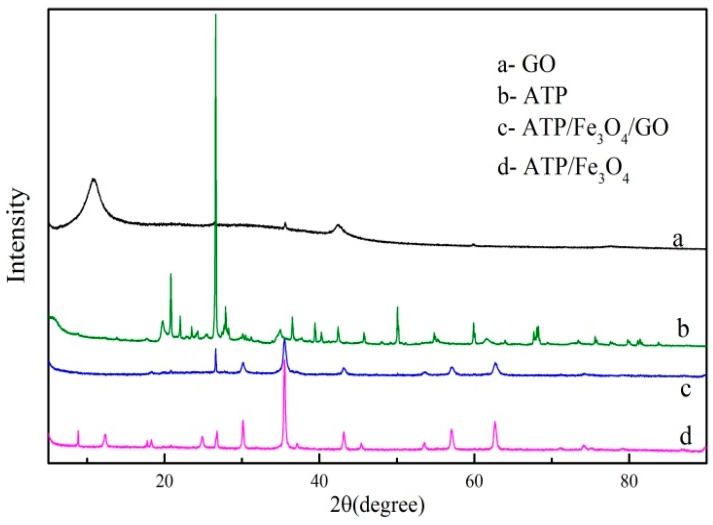
The X-ray diffraction patterns of GO, ATP, ATP/Fe_3_O_4_, and ATP/Fe_3_O_4_/GO.

**Figure 4 materials-12-00924-f004:**
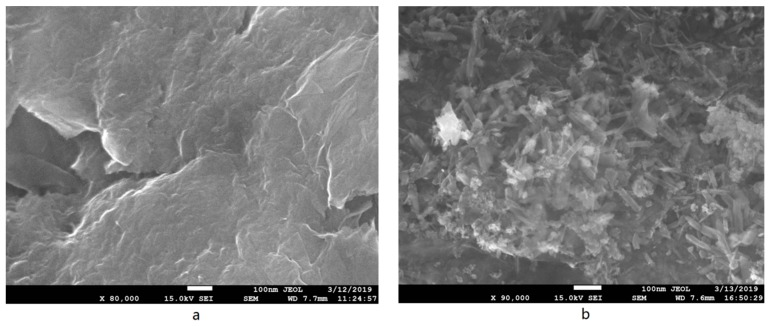
SEM micrographs of GO (**a**) and ATP/Fe_3_O_4_/GO (**b**).

**Figure 5 materials-12-00924-f005:**
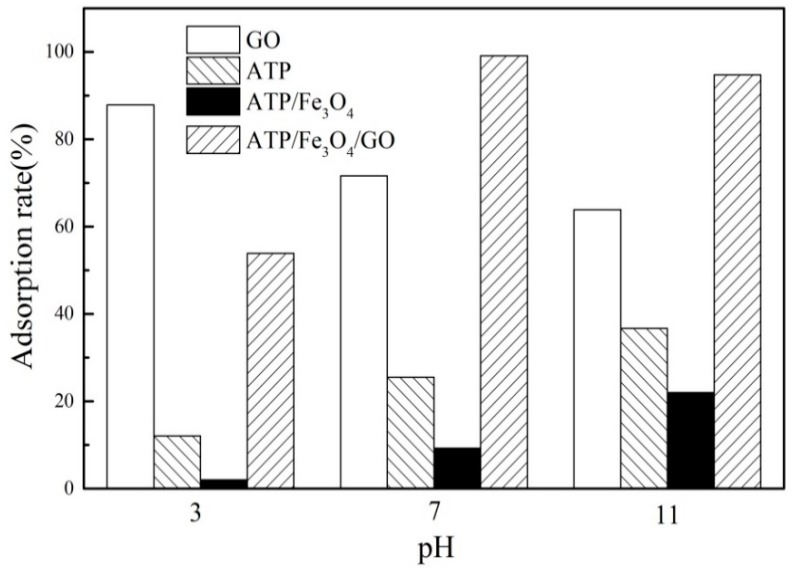
A comparison of the adsorption capacities of propranolol (PRO) onto GO, ATP, ATP/Fe_3_O_4_, and ATP/Fe_3_O_4_/GO.

**Figure 6 materials-12-00924-f006:**
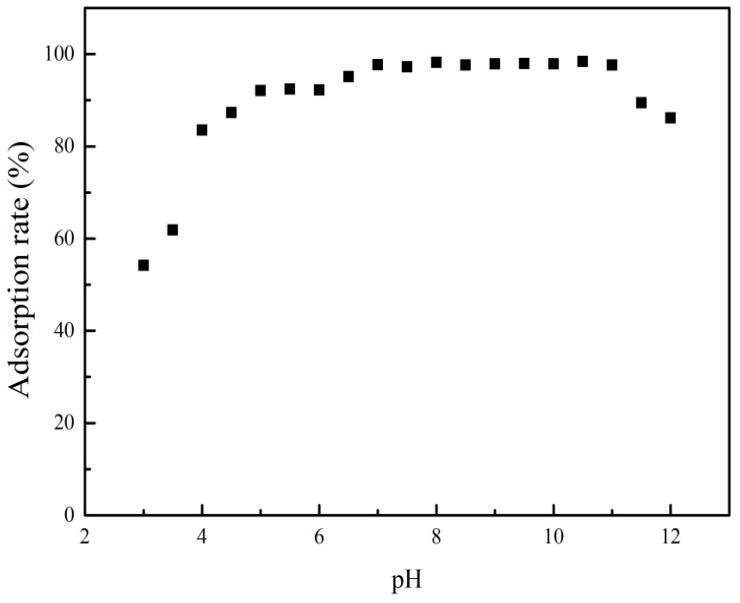
The effect of pH on the adsorption of PRO onto ATP/Fe_3_O_4_/GO.

**Figure 7 materials-12-00924-f007:**
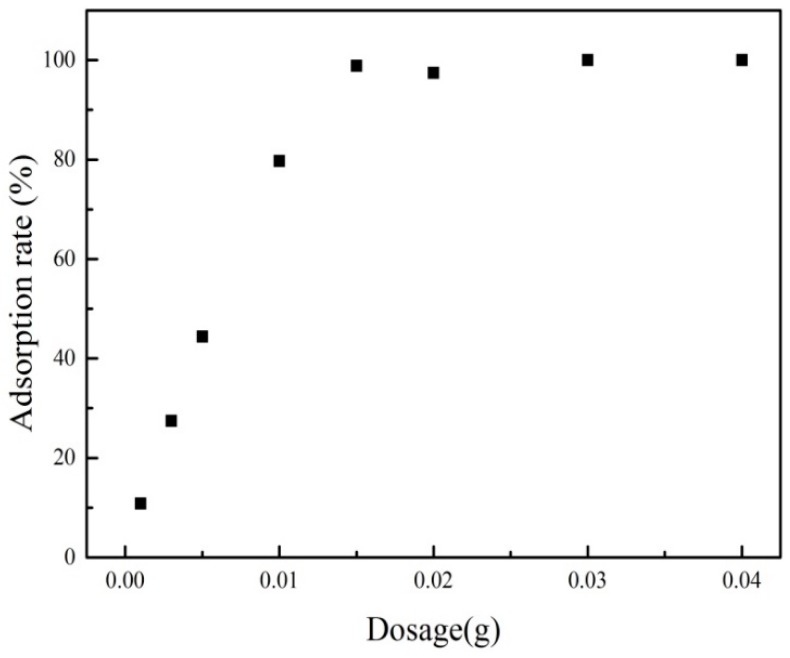
The effect of adsorbent mass on the adsorption of PRO onto ATP/Fe_3_O_4_/GO.

**Figure 8 materials-12-00924-f008:**
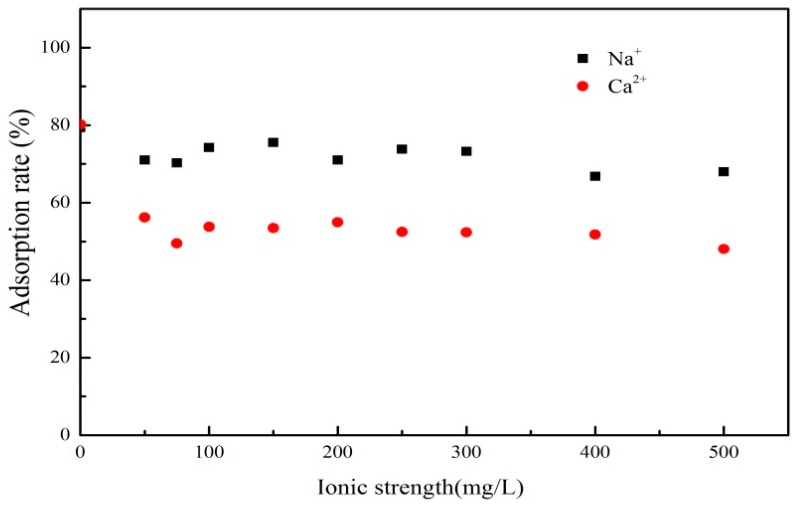
The effect of ionic strength on the adsorption of PRO onto ATP/Fe_3_O_4_/GO.

**Figure 9 materials-12-00924-f009:**
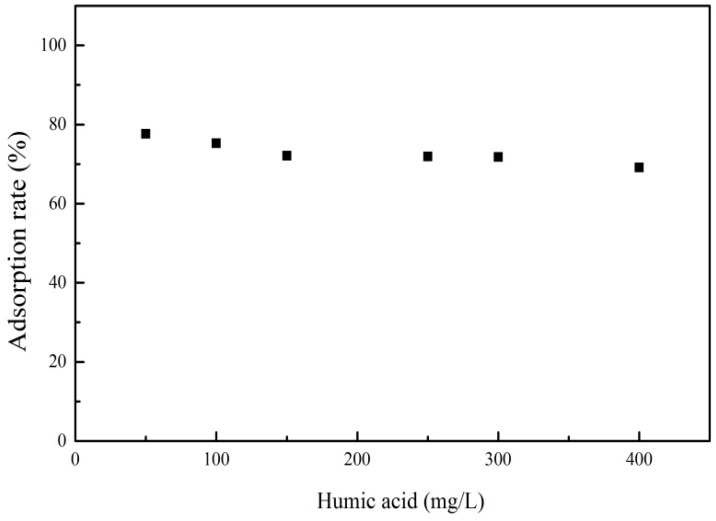
The effect of humic acid on the adsorption of PRO onto ATP/Fe_3_O_4_/GO.

**Figure 10 materials-12-00924-f010:**
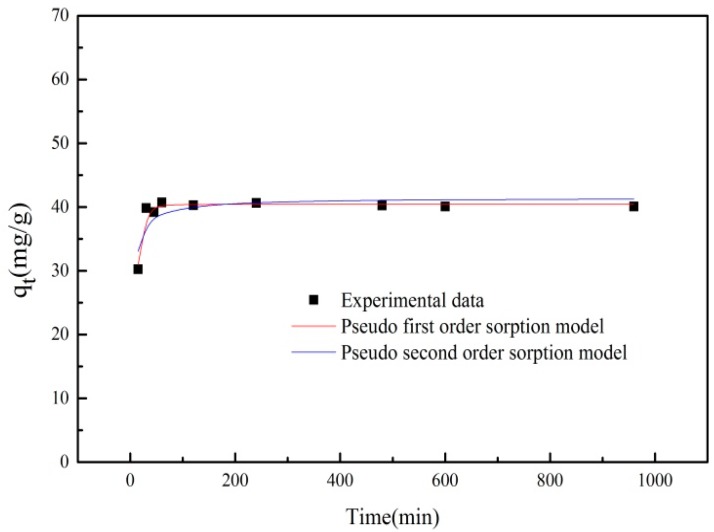
The adsorption kinetics of PRO onto ATP/Fe_3_O_4_/GO.

**Figure 11 materials-12-00924-f011:**
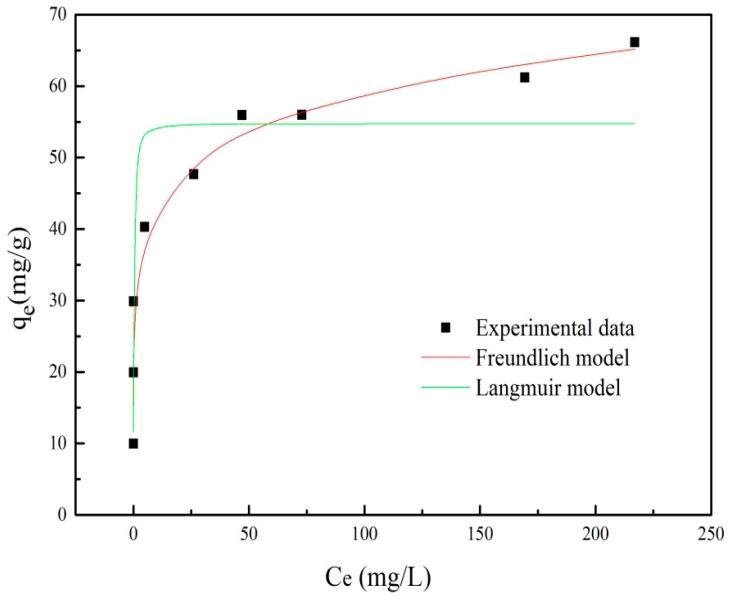
The adsorption isotherm of PRO onto ATP/Fe_3_O_4_/GO.

**Table 1 materials-12-00924-t001:** The kinetic parameters for PRO adsorption onto ATP/Fe_3_O_4_/GO.

q_exp_ (mg/g)	Pseudo-First-Order Equation			Pseudo-Second-Order Equation		
	*K*_1_ (1/min)	*q_cal_* (mg/g)	*R* ^2^	*K*_2_ (g/(mg·min))	*q_cal_* (mg/g)	*R* ^2^
42.07	0.097	40.432	0.947	0.0064	41.415	0.682

**Table 2 materials-12-00924-t002:** The adsorption isotherms parameters for PRO adsorption onto ATP/Fe_3_O_4_/GO.

Langmuir Equation			Freundlich Equation		
*q_m_* (mg/g)	*K_L_* (L/mg)	*R* ^2^	*K_F_* (mg/g)	*n*	*R* ^2^
54.763	21.747	0.841	32.361	7.683	0.945

**Table 3 materials-12-00924-t003:** The thermodynamic parameters for PRO adsorption onto ATP/Fe_3_O_4_/GO.

C_0_ (mg/L)	ΔH (kJ/mol)	ΔS (J/(mol·K))	ΔG (kJ/mol)		
			298 (K)	308 (K)	318 (K)
50	12.22	45.72	−1.41	−1.87	−2.33

**Table 4 materials-12-00924-t004:** The adsorption capacity for PRO onto different adsorbents.

Adsorption	Adsorption Amount	Reference
ATP/Fe_3_O_4_/GO	46.8	This work
Acid-activated ATP	48.05	[22]
Chitosan-ATP	26.38	[22]
KH550-ATP	24.56	[22]
Powder activated carbon	46.78	[22]

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
