# Peer review of "Fast Removal of Propranolol from Water by Attapulgite/Graphene Oxide Magnetic Ternary Composites"

_materials, 2019, doi:10.3390/ma12060924_

Reviewer 1 Report

1. Figure quailed was poor, the author should increase the figure quailed.

2. The author should add ATP/Fe3O4/GO morphology.

3. PRO loading capacity of ATP/Fe3O4/GO should be compared with some high loading adsorbents.

4. Add Thermodynamic study.

5. Add the physical and chemical characterization of PRO loaded ATP/Fe3O4/GO and discussed in detailed.

Author Response

Response to Reviewer 1 Comments

Point 1: Figure quailed was poor, the author should increase the figure quailed.

Response 1: Thank you very much for reminding us of the figures quality problem. We apologize for our mistakes in the original figures. All the low quality figures were replaced and new high quality figures were added so that everyone reading the paper could clearly understand the figures.

Point 2: The author should add ATP/Fe3O4/GO morphology.

Response 2: Thank you very much for your valuable suggestion, which can help us to improve the quality of this paper. In revised manuscript, SEM characterization was used to describe ATP/Fe3O4/GO morphology. We added a total of two new figures. Figure 4(a) was SEM micrographs of GO and Figure 4(b) was SEM micrographs of ATP/Fe3O4/GO. The detail was explained in Section 3.1.4. The added sentences are at lines 180-189: “SEM was used to characterize the synthesized GO and ATP/Fe3O4/GO. As it can be seen in Figure 4 (a), GO formed the sheet-like structure and had a smooth surface without holes onto its surface. Partial agglomeration was also observed. The smooth surface of GO became coarse and dense after the loading of ATP and Fe3O4 particles (Figure 4 (b)). Many rod-shaped ATPs were crosslinked on the surface of the GO, indicating that GO acted as a dispersant. Fe3O4 particles were also observed. They were agglomerated on the surface of GO and ATP, and the results were consistent with FTIR and XRD characterization. It was speculated that an increasing in surface area of ternary composites than that of GO is reasonable, and its adsorption capacity was discussed in subsequent chapters.”

Point 3: PRO loading capacity of ATP/Fe3O4/GO should be compared with some high loading adsorbents.

Response 3: Thank you very much for your helpful and valuable comments. We added this section in Section 3.7 in revised manuscript. We compared the adsorption capacity of ATP/Fe3O4/GO with other adsorbents including acid-activated ATP, chitosan-ATP, KH550-ATP, and powder activated carbon in our previous work. And we showed the related data in Table 4.

Point 4: Add Thermodynamic study.

Response 4: Thank you very much for your thoughtful and precise comment. In revised paper, we tested PRO removal rate in three different temperatures, calculated the relevant thermodynamic parameters. All information was showed in Section 3.6.

Point 5: Add the physical and chemical characterization of PRO loaded ATP/Fe3O4/GO and discussed in detailed.

Response 5: Thank you very much for your good comment. We are very sorry for our unclear description of the physical and chemical characterization of PRO loaded ATP/Fe3O4/GO. Therefore, we further explained this part in the revised manuscript. For FTIR spectral, we added the interpretation of the interaction among ATP, GO and Fe3O4. The added sentences are at lines 146-154: “The interaction among GO, ATP, and Fe3O4 was affirmed by the change of the peaks intensity at about 3432 cm-1, 1721 cm-1 and 1222 cm-1, and these peaks of ATP/Fe3O4/GO were weaker than those of GO, but stronger than those of ATP/Fe3O4. The peaks at 1721 cm-1 and 1222 cm-1 of GO were assigned to C=O stretching in carboxylic acid and C-O stretching in epoxy, respectively. However, ATP had no obvious characteristic peaks at peak positions, leading to weakening of the peak intensity of the ternary complex at the above peak positions. The change in the peak intensity at 3432 cm-1 may be ascribed to the overlap of the O–H stretching vibrations of GO and the bend vibration of the zeolite water of ATP.” For zeta potential, we added a sentence “The surface charge of the adsorbents was affected by solution pH to a large extent.” at the beginning of the paragraph, rewrote some sentences that were not properly expressed. For example, the sentence “Magnetic materials showed a higher zeta potential than raw ATP, which suggesting Fe3O4 played an important role in the surface properties of materials.” was change to “Fe3O4 and ATP/Fe3O4 showed a higher zeta potential than three other materials, which suggesting Fe3O4 particles may change the surface charge of materials.”. And as you suggested, we also added SEM micrographs of GO and ATP/Fe3O4/GO in Section 3.1.4.

Reviewer 2 Report

The authors have synthetized and characterized and interesting composite based on a natural clay. They studied its performance on the removal of a pharmaceutical compound usually found in wastewater under different scenarios, considering pH, sorbent mass, ionic strength and humic acid concentration. The experiments are consistent and the authors point out the retention mechanisms of the PRO molecules on the composite.

Figures 5, 6, 7, 8 and 9 should be presented in a most comparable way, using the same y-axis scale in each one. Thus, the point that in common for every scenario would be clearly identified.

According to section 2.3, the batch experiments were conducted using 25 mL of a PRO solution of 20 mg /L and 0.01 g of sorbent. Thus, maximum PRO uptake is 50 mg/g. Have you considered to report the results in relative terms. i.e. removal efficiency?

In section 3.3.2, reporting the uptake in absolute terms difficult the interpretation of the results, since it it quite obvious that increasing the sorbent mass leads to a lower uptake. But, considering that it is not a lineal behavior, it may be more interesting to express the effect of the sorbent dosage in terms of removal efficiency.

Figures 9 and 10: in order to clarify the expression of the results, the data corresponding to models adjustment should be plotted using only lines, and experimental data using dots.

There is not a general discussion of the results pointing out the applicability of the material in real scenarios. Authors should indicate clearly what is the relevance of the different scenarios investigated. 

Author Response

Response to Reviewer 2 Comments

Point 1: Figures 5, 6, 7, 8 and 9 should be presented in a most comparable way, using the same y-axis scale in each one. Thus, the point that in common for every scenario would be clearly identified.

Response 1: Thank you very much for your valuable suggestion. Due to the addition of the SEM micrographs of GO and ATP/Fe3O4/GO, the number of original Figures 5, 6, 7, 8 and 9 changed to Figures 6, 7, 8, 9, and 10. We changed the y-axis label adsorption capacity (mg/g) of Figure 6, 7, 8, and 9 to the adsorption rate (%). And the range of the y-axis is 0-110. For Figure 10 about adsorption kinetic, we considered that kinetic data needed to fit the kinetic equations, and if the ordinate of the picture was changed to the adsorption rate, the fitting trend cannot be directly seen from the figure. So we have not changed the ordinate of Figure 10.

Point 2: According to section 2.3, the batch experiments were conducted using 25 mL of a PRO solution of 20 mg /L and 0.01 g of sorbent. Thus, maximum PRO uptake is 50 mg/g. Have you considered to report the results in relative terms. i.e. removal efficiency?

Response 2: Thank you very much for your good comment. Based on your suggestion, we changed y-axis of Figure 6, 7, 8, and 9 from adsorption capacity (mg/g) to adsorption rate (%), and recalculated relevant data.

Point 3: In section 3.3.2, reporting the uptake in absolute terms difficult the interpretation of the results, since it it quite obvious that increasing the sorbent mass leads to a lower uptake. But, considering that it is not a lineal behavior, it may be more interesting to express the effect of the sorbent dosage in terms of removal efficiency.

Response 3: Thank you very much for giving us so many helpful and valueable suggestions. We changed the y-axis from adsorption capacity (mg/g) to adsorption rate (%) in Figure 7.

Point 4: Figures 9 and 10: in order to clarify the expression of the results, the data corresponding to models adjustment should be plotted using only lines, and experimental data using dots.

Response 4: Thank you very much for reminding us the improper exhibition in Figures 9 and 10.As you suggested, we use the dots to plot the experimental data, lines to the data corresponding to models adjustment so that we can obtain the trend of data fitting more clearly.

Point 5: There is not a general discussion of the results pointing out the applicability of the material in real scenarios. Authors should indicate clearly what is the relevance of the different scenarios investigated.

Response 5: Thank you very much for your helpful comments. Your suggestion makes a lot of sense for our follow-up work. The materials we are currently working on are only tested in the laboratory. This material was found to be useful in water bodies of different pH. The results provides a theoretical basis for removing PRO in water. However, the PRO removal by this material in real water has not been tested. In this article we only studied the feasibility of removing PRO by this material from water.

Round  2

Reviewer 1 Report

In the manuscript the authors described the adsorption removal of PRO. The detailed chemical characterizations of  the absorbent materials have been carried out. Effect of different parameter such as pH, contact time, metal concentration, absorbent concentration on the efficiency of absorption has also been discussed in detailed. The manuscript is recommended for accepted.

Reviewer 2 Report

After revision, the authors have significantly improved the manuscript quality.